# Conversion of Lignocellulose for Bioethanol Production, Applied in Bio-Polyethylene Terephthalate

**DOI:** 10.3390/polym13172886

**Published:** 2021-08-27

**Authors:** Damayanti Damayanti, Didik Supriyadi, Devita Amelia, Desi Riana Saputri, Yuniar Luthfia Listya Devi, Wika Atro Auriyani, Ho Shing Wu

**Affiliations:** 1Department of Chemical Engineering and Materials Science, Yuan Ze University, 135 Yuan-Tung Road, Chung-Li, Taoyuan 32003, Taiwan; damayanti@tk.itera.ac.id; 2Department of Chemical Engineering, Institut Teknologi Sumatera, Jl. Terusan Ryacudu, Way Huwi, Kec. Jati Agung, Lampung Selatan, Lampung 35365, Indonesia; didik.supriyadi@tk.itera.ac.id (D.S.); devita.amelia@tk.itera.ac.id (D.A.); riana.saputri@tk.itera.ac.id (D.R.S.); yuniar.listyadevi@tk.itera.ac.id (Y.L.L.D.); atro.auriyani@tk.itera.ac.id (W.A.A.)

**Keywords:** bioethanol, bio-polyethylene terephthalate, pretreatment, hydrolysis, fermentation

## Abstract

The increasing demand for petroleum-based polyethylene terephthalate (PET) grows population impacts daily. A greener and more sustainable raw material, lignocellulose, is a promising replacement of petroleum-based raw materials to convert into bio-PET. This paper reviews the recent development of lignocellulose conversion into bio-PET through bioethanol reaction pathways. This review addresses lignocellulose properties, bioethanol production processes, separation processes of bioethanol, and the production of bio-terephthalic acid and bio-polyethylene terephthalate. The article also discusses the current industries that manufacture alcohol-based raw materials for bio-PET or bio-PET products. In the future, the production of bio-PET from biomass will increase due to the scarcity of petroleum-based raw materials.

## 1. Introduction

Polyethylene terephthalate (PET) is primarily used as a raw material in the production of synthetic fibers (65%) and packaging (35%) [1]. PET is produced by polycondensing petroleum-based pure terephthalic acid (TPA) and ethylene glycol (EG) [2]. Because PET has excellent mechanical properties, the demand for PET-based products (textile, food packaging, and bottles) is very high. For instance, the total production of PET in 2015 as packaging was estimated at 18.8 million tons [3,4]. However, the strong demand for PET has a negative impact on the environment because PET products are difficult to degrade in the soil [5].

Aside from the environmental concern, the use of non-renewable materials in the production of TPA and EG is a critical issue that must be addressed for the future of PET production. Furthermore, replacing petroleum-based raw materials with more sustainable materials is essential. Lignocellulose is a promising candidate for replacing conventional raw materials of bio-PET production. The advantage of utilizing lignocellulose, especially lignocellulose from agriculture or forestry waste, is reducing the waste in the environment and increasing the value of the waste [6]. Before utilizing lignocellulose as bio-PET’s raw material, lignocellulose must be converted into alcohol products (bioethanol, biobutanol) and react to TPA and EG. Some studies have conducted review papers to discuss about producing bio-PET from renewable material [1,7]; however, there are a few papers to review about detail production bio-PET from downstream to upstream process.

There are several steps to produce bioethanol from lignocellulose, such as pretreatment, hydrolysis, and fermentation. The pretreatment stage is the most expensive and limiting step on lignocellulose conversion because the pretreatment process takes 20% of the total cost production [8,9]. Nevertheless, the pretreatment stage can be optimized by increasing the concentration of fermentable sugar compounds followed by enzymatic saccharification. Thus, the overall efficiency of the process can be improved [10]. The physical, chemical, physical-chemical technologies, and biological are applied through the pretreatment stage. This process aims to separate main components in lignocellulose into cellulose, hemicellulose, and lignin. 

The physical pretreatment is conducted to reduce the size of lignocellulose particle, degree of polymerization, and crystallinity of lignocellulose and enlarge the surface area of the particle [11]. This method consists of milling, microwave, extrusion, and ultrasonic. Higher energy consumption is an essential factor that must be considered when choosing this method [9]. On the other hand, the chemical pretreatment methods include alkaline hydrolysis, acid hydrolysis, organosolv process, ionic liquid, and deep eutectic solvents [9]. These methods will deteriorate the structure of lignocellulose with the chemical substances in the absence of pressure [12]. Physical-chemical technology is the hybrid pretreatment between physical and chemical methods that combines the pressure or temperature in the physical process and chemical substances in the chemical process. This method consists of CO_2_ explosion, steam explosion, ammonia fiber explosion, and liquid hot water [9,12]. The last pretreatment method that is usually used for handling lignocellulose is biological treatment. This treatment uses bacteria, fungi, and enzymes to decompose lignin [9]. The fermentation process is applied to produce bioethanol from lignocellulose. The various bacteria, yeast, and fungi are developed for lignocellulosic fermentation, e.g., *S. cerevisiae* can get higher conversion and bioethanol yield [13,14]. Furthermore, bioethanol is conducted catalytically dehydrated to ethylene [15]. It could be polymerized to be polyethylene and oxidized to ethylene oxide [16].

The study considers the importance of lignocellulose as a sustainable raw material via bioethanol pathway to produce bio-PET in the future. This article reviews lignocellulose characteristics, a distinct way of converting lignocellulose into bioethanol, bioethanol separation, and converting bio-alcohol products into bio-PET. Finally, the recent industrial scale of reactants for making bio-PET or bio-PET reveals the development of bio-PET in the world.

## 2. Properties of Lignocellulose

The lignocellulosic biomass compounds are divided into several polymers, for instance, cellulose, hemicellulose, and lignin [17]. The structure of lignocellulose polymers is very complicated. Nevertheless, there are more than 200 value-added chemicals [6]. It may be obtained from lignocellulose biomass with different types of treatment methods. The sugar compounds C_5_ (arabinose, xylose) and C_6_ (rhamnose, mannose, rhamnose) can be converted into glycerol xylitol/arabinitol, 1,4-diacids (succinic acid, malic and fumaric acid), 3-hydroxy propionic acid (3-HPA), 2,5-furan dicarboxylic acid (2,5-FDCA), glutamic acid, levulinic acid, itaconic acid, 3-hydroxybutyrolactone (3-HBL), and others [6,18]. Biomass is transformed by using biochemical and thermochemical processes. Heat and catalysts are applied in thermochemical processes [19]. Nevertheless, enzymes or microorganisms are utilizing in biochemical processes [20]. Cellulose is a natural substance that is the primary constituent of plant cell walls [21], with the range of diameters from 2 to 20 nm and lengths ranging from 100 to 40,000 nm [22]. The cellulose structure comprises hydroxyl groups (OH^−^) combined into a cyclic compound with a configured hexagon shape and interconnected by an oxygen bridge, as shown in Figure 1. The oxygen bridge is a β-1,4-glucosidic bond that binds cellulose units together. Hydrogen bonds are commonly present in cellulose molecules.

The composition of hemicellulose differs. Hemicellulose is the second most abundant polymer, with about 20–50% lignocellulose [23]. Furthermore, the structure of hemicellulose polymer is a heteropolymer. It had an amorphous random structure with hardly any flexibility. Unlike linear biopolymers made up of glucose monomers. Hemicellulose is a branching polysaccharide made up of sugars. The several sugar compounds include arabinose, xylose, fucose, and galactose. Several linkages of monomers connect these kinds of sugars [24]. This kind of hydroxyl group can produce well-ordered hydrogen linkages, resulting in a solid crystalline structure.

Furthermore, the partial cellulose linkages are arrayed randomly, resulting in the amorphous cellulose region [25]. Figure 2 shows that the chemical models of sugar in hemicellulose. The cellulose, hemicellulose, and lignin are broken down by acid or the enzymatic hydrolysis process. Furthermore, xylose is a pentose sugar that can be dehydrated to produce furfural. In comparison, the organic acid compounds, such as formic and acetic acid, are formed via the hydrolysis process [26].

Lignin is a polymer that tightens and stiffens the cell walls. The main lignin monomers are aromatic alcohol (guaiacyl (G), p-hydroxyphenyl (H), and syringyl (S)), derived from coniferyl, p-coumaryl, and sinapyl alcohol. [27,28]. In addition, lignin and hemicellulose contribute to cellulose’s cementing stages. Plant cell walls can indeed be described as a series of elastic fibers of spirally oriented cellulose microfibers crosslinked by hemicellulose and lignin molecules. In contrast to cellulose and hemicelluloses, which are made up of carbohydrate units [29].

Lignin has a diversity of C-O or C-C bond types with varying quantities formed in the crosslinking reactions that occur in lignin biosynthesis, such as biphenyl (5–5′), resinol (-5), aryglycerol-ether dimer (α-O-4 and β-O-4), phenylcoumaran (β-β), diphenylethane (-1), siaryl ether (4-O-5), and spirodienon. Lignin is predicted to play a significant role in the sustainable development of aromatic compounds, monomers, or building blocks, for instance, benzene, xylene, toluene, phenol, and vanillin, in contrast to its effective uses as a polymer [27]. 

## 3. Conversion of Lignocellulose into Biochemical Substance

Several agricultural residues, such as sorghum husk, rice straw, bagasse, corn husk, corn stover, maize forage, wheat straw, coffee, and orange peel, have been investigated for biofuel production [30]. Several methods and technologies are applied to chemical and biochemical routes to degrade lignocellulose to fuels and chemical building blocks (ethanol, butanol, phenols, organic compound, and alcohol). Furthermore, the chemical blocks are commonly used in gasoline and diesel mixtures. Bioethanol production through several processes, such as pretreatment, hydrolysis, and fermentation, was conducted. The complex polymer of lignocellulose can be converted from polysaccharides into fermentable sugars. A variety of pretreatment procedures are available [31]. Figure 3 relates to the schematic hydrolysis of lignocellulose to ethanol formation. The beginning process is the pretreatment of ionic liquid, microwave, dilute acid, or steam explosion, following hydrolysis and fermentation. 

The pretreatment method can help break down lignin and glycosidic chains by minimizing structural crystallization and increasing lignocellulose digestibility [32,33]. Due to the inherent recalcitrance of biomass, a crucial step known as biomass pretreatment is needed before the hydrolysis process. Commonly, pretreatment of lignocellulose biomass is performed to remove or redistribute plant cell wall parts, break up the molecule carbohydrate-lignin chain, and finally raise the cellulose accessibility to enzymes [33]. The concentration of fermentable sugars can be increased with proper pretreatment. An effective pretreatment procedure eliminates the requirement for biomass size reduction, allows lignocellulosic biomass to be hydrolyzed quickly, and produces a higher monomeric sugar yield. The synthesis of inhibitory compounds should be limited, and energy needs and capital and operating costs should be reduced [34]. The different types of pretreatment processes would be affected by yields and products [35,36]. Table 1 shows that the comparison of various pretreatment methods with energy consumption. Acid chemicals can approach the chemical pretreatment to break the macromolecules of lignocellulose polymer. These technologies are simple to operate and produce high yields in a short amount of time. Furthermore, acid pretreatment is an approach for obtaining large yields of sugars from lignocellulosic biomass [37]. On the other hand, enzymatic hydrolysis has various advantages. For instance, the formation of unwanted byproducts decreased, low acid waste was not required for corrosion-resistant apparatus, and the reaction conversion was nearly complete [38]. 

### 3.1. Mechanical Pretreatment 

Commonly, mechanical pretreatment consists of simple procedures and a lack of environmental issues. Nevertheless, this approach consumes a lot of power and energy, resulting in higher manufacturing expenses. The several techniques adapted include milling, chipping, and grinding [39]. The mechanical pretreatment is required to change the biomass framework to disrupt the crystalline structure of cellulose, hence increasing the biomass’s surface area for further processing. In addition, mechanical pretreatment can decrease the hydrolysis time. The enzymes may have easier access to lignocellulose regarding the small particle size of biomass [40]. Hammer-milling is applied for pulverization of lignocellulose because of the significant reduction ratio and simple adjusting of the range of particle size. The particle size reduction can be operated as coarse grinding [41]. Bai et al. studied mechanical pretreatment via rod-milling with a rod-milling time of 30–240 min. It can minimize the size of particles and cellulose crystallinity and enhance wheat straw’s specific surface area and pore volume [42]. In addition, Gu et al. investigated the effect of the planetary ball mill with pre-milled wood fiber. The crystalline cellulose was decreased significantly started from 40.7% to 11.7%. The yield range between glucose and xylose/mannose by ball mill was 24.4–59.6% and 11.9–23.8%, respectively [40]. 

### 3.2. Chemical Pretreatment

The organic acids, such as formic acid and acetic acid, were used in the chemical pretreatments. The acidic pretreatment generates sugar compounds, followed by raising the toxic inhibitor formation in the chemical reaction mixture, while alkaline pretreatment results in significant delignification and minimal solid recovery [43]. The ozonolysis is applied for oxidative pretreatment of lignocellulose with more reactivity; nevertheless, there is low selectivity of the substrate. Travaini et al. investigated ozonolysis with different raw materials, such as wood chips, pulps, bagasse, grass, and microalgae. The ozonolysis under extreme oxidative power of ozone gas (O_3_) in the presence of water to produce the hydroxyl radical (OH^−^) enhanced reactivity [44,45]. Furthermore, the acetyl group can be prohibited from the iteration between enzymes and cellulose. Acetyl groups replace the hydroxyl groups of cellulose to link cellulase, and cellulose cannot produce hydrogen bonds, and this can reduce the enzymatic hydrolysis of the substrates [46]. 

### 3.3. Microwave Irradiation Pretreatment

Microwaves are common forms of electromagnetic radiation that appear as light and in the form of waves. Microwaves are primarily non-ionizing, while low-frequency waves are non-ionizing. Microwaves are produced by reversing the dipole, separating two equal-sized charges separated by a given distance. The positive charge then transfers to the bottom, while the negative charge moves opposite [47]. The range wavelength’s location of microwave non-ionizing electromagnetic radiation is 3 × 10^2^–3 × 10^5^ Mhz on the electromagnetic spectrum [48]. There are two methods of lignocellulosic pretreatment with a catalyst: (1) microwave-assisted solvolysis with the medium temperature reaction below 200 °C and (2) the high temperature of a reaction more than 400 °C by microwave-assisted pyrolysis. Furthermore, there are some benefits of microwave irradiation pretreatment over the conventional heating method: (1) decrease reaction time and increase heat transfer rate, (2) heating performance that is constant over the volume, (3) high energy efficiency to reduce operation cost, and (4) undesired formation suppressed due to little degradation. Moreover, the hydrothermal pretreatment under microwave irradiation can be removed from acetyl groups in the present hemicellulose [49].

**Table 1 polymers-13-02886-t001:** Comparison of various pretreatment methods with energy consumption.

Feedstocks	Type of Pretreatment	Promotor for Bioethanol	Yield Bioethanol, %	Total Selling Price (US$/Kg Bioethanol)	Energy Consumption (MJ/Kg Bioethanol)	Reference
Agricultural residues	Organosolv	*Zymomonas mobilis*	NA	1.27	331.27	[50]
Beet molasses	NA	*Saccharomyces cerevisiae*	49	NA	227.64	[51]
Waste bamboo stems	Hydrogen peroxide/acetic acid (HPAC)	*S. cerevisiae KCTC 7906*	83.1	1.2	265.3	[52]
Bagasse	Acid (H_2_SO_4_)	*Saccharomyces cerevisiae*	638 (mg/L)	NA	29	[53]
Olive pruning debris	Steam explosion	*Saccharomyces cerevisiae (SIGMA II Type)*	95	NA	8.53	[54]
Olive pruning debris	Grinding	*Saccharomyces cerevisiae (SIGMA II Type)*	11	NA	180	[54]
Olive pruning debris	Torrefaction & grinding	*Saccharomyces cerevisiae (SIGMA II Type)*	10	NA	64.29	[54]

### 3.4. Biological Pretreatment

Microorganisms and their enzyme systems are used in biological pretreatment to modify high-molecular-weight compounds of lignocellulose structure. The biological pretreatment can be conducted under mild conditions, low energy consumption, and produced less toxic substances [55]. In the biological pretreatment, many fungi and bacteria produced ligninolytic, cellulolytic, lipolytic, proteolytic, pectinolytic, and amylolytic enzymes [20]. On the other hand, the disadvantage of biological pretreatment is the low reaction rate and long reaction time to modify the structure of lignocellulose. Furthermore, the increasing reaction temperature would decrease the total number of microorganisms. Anaerobic digestion is used to enhance the biofuel yield via ultrasonication [56]. 

*Bacillus Firmus K-1* is the type of xylanolytic bacterium. It generates extracellular xylanolytic enzymes, such as *xylanases, β-xylosidases, α-L-arabinofuranosidases,* and *acetyl esterases*, without cellulase activity. In addition, the microorganism grows very quickly in the minimum medium, using husk and rice straw and a low-cost carbon source for microbial growth and enzyme production [57]. 

### 3.5. Ionic Liquid and Deep Eutectic Solvent Pretreatment

Low-cost ionic liquid (IL) is a promising method for the lignocellulosic biomass pretreatment to produce bioethanol. Ziaei-Rad et al. investigated utilizing low-cost IL, [TEA][HSO_4_], for wheat straw pretreatment. The pretreatment was investigated in conditions characterized by 130 °C, a high solid-to-solvent load ratio of 1:5 g/g, and 20 wt% water to separate lignin and carbohydrates from the source. The highest delignification rate of 80% and the hemicellulose removal rate of 64.45% were observed in the case of a 3-h pretreated sample with [TEA][HSO_4_] [58]. 

Kulshrestha et al. reported that the saccharification yield was 95.5%, using ethyl ammonium nitrate (IL), whereas, among hydrophobic deep eutectic solvent (DES), menthol: lactic acid exhibited the saccharification yield of 85.7%, which did not require any additional high temperature or other pretreatments for hydrolysis. This result is indicated as the potential solvent system to reduce the cost of bioethanol production concerning the harsh conditions in a conventional method. Their results indicated that the identified IL and DES could be used as a green and sustainable alternative for the pretreatment of biomass for biofuel production [59]. 

## 4. Lignocellulose Hydrolysis

The production of bio-based chemical compounds and sugar by using lignocellulosic hydrolysis is a crucial stage. Lignocellulose conversion is included in the following processes. Cellulose lignocellulosic materials are hydrolyzed to fermentable reducing sugars and are then fermented to ethanol. The hydrolysis process is catalyzed via cellulase enzymes, and the fermentation process is conducted by bacteria or yeast. Some factors would be affected in the hydrolysis process: porosity of materials and the crystallinity of cellulose fiber, lignin, and hemicellulose content. Cellulase enzymes find it challenging to reach cellulose because of hemicellulose and lignin, which are affected by lowering hydrolysis efficiency [60,61]. 

The advantages and disadvantages of enzymatic and acid hydrolysis are shown in Table 2. The enzymatic hydrolysis had some benefits. For instance, this process is not related to the toxic production process; therefore, it could save the cost of detoxification inhibitors. Furthermore, a high substrate concentration allows for a higher yield at the expense of increased enzyme use [62,63]. On the other hand, cellulose and hemicellulose could be broken down by hydrolysis under dilute acid. Sugar compounds (glucose, xylose, mannose, and galactose) and organic acids (acetic and formic acid) are the most common results of hemicellulose hydrolysis. During acidic hydrolysis, cellulose could be hydrolyzed to glucose form, and then the glucose can be converted into high-value compounds, such as hydroxymethylfurfural (HMF) [64].

### 4.1. Enzymatic Hydrolysis

The depolymerization of cellulose into glucose can be conducted by enzymatic hydrolysis. The enzymatic hydrolysis is operated under mild conditions; for instance, the range temperature and pH are 40–50 °C and 4.5–5, respectively [31]. The cellulose hydrolysis process can be broken up into two basic enzymatic schemes, multi-enzymes, bacteria, and fungi applied the free extracellular enzymes. Furthermore, multiple enzymatic activities are essential to hydrolyze cellulose through soluble monosaccharides, and they can be implemented by assimilated cells [65]. Moreover, the process of enzymatic hydrolysis was discovered by Yang et al.; some of the methods are needed in heterogeneous reactions. The insoluble cellulose was broken down through a solid-liquid interface by the end of cellobiohydrolases/exoglucanases. The beginning reaction was followed by liquid-phase hydrolysis of soluble intermediates, and the catalytically cleaved was applied to produce glucose by β-glucosidase [66].

**Table 2 polymers-13-02886-t002:** The advantages and disadvantages of various types of hydrolysis.

Type of Hydrolysis	Biological/Chemical Agent Used	Concentration (%)	Yield Glucose (%)	Advantages	Disadvantages	Reference
Diluted Acid	H_2_SO_4_	10	NA	High efficiency of sugar recovery Low pressures and temperatures Low-cost materials	A relatively slow rate of conversion The conversion rate is relatively slow To neutralize, a sufficient amount of lime must be applied, and expensive	[67]
Enzymatic	*A. cellulolyticus*	22.5–90.0 mg	≥65	High glucose yield The increased specific activity of cellulase Higher hydrolyzing activity	Long reaction time	[68]
Enzymatic	*T. reesei*	22.5–90.0 mg	≤60	Greater activity in hydrolyzing xylan Low-temperature reaction	Long reaction time	[68]
Acid	HCl	NA	NA	Hydrochloric acid hydrolysis has a higher efficiency than sulfuric acid hydrolysis.	Waste treatment is more complicated than sulfuric acid treatment. The cost of hydrochloric acid is higher than sulfuric acid, corrosion-resistant equipment is required	[69]
Diluted acid	H_3_PO_4_	2.5	90	At mild temperatures, a high-rate constant Minimize sugar degradation to unwanted by-products	The high expense of concentrated acid A recovery stage is required to reduce the cost	[70,71]
Diluted Acid	H_2_SO_4_	20–33	78–82	Higher sugar yield	Long reaction time High acid concentrations will pollute the ecosystem	[69]
Enzymatic	*Cellulase*	NA	70	Environmentally friendly Low cost Degrade cellulose into small-chain polysaccharides Not causing a corrosion issue	The enzymes are relatively pricey	[72,73]

Song et al. investigated the effect of sequential fermentation by *Saccharomyces cerevisiae* for bioethanol production. The bioethanol theoretical yield was up to 81%. The total material costs in bioethanol production (biomass, H_2_O_2_, acetic acid, enzyme, chemical fermentation, and yeast) by conventional and sequential were 3.36 and 2.69 $/kg bioethanol, respectively. Furthermore, the total energy consumed (enzymatic hydrolysis, glucose fermentation, pervaporation, and xylose fermentation) for conventional and sequential was 99.4 (1.8) and 152.2 (1.9), MJ ($/kg bioethanol), respectively [74]. 

Enzymatic hydrolysis is conducted in a specific section; hemicellulase enzymes are only responsible for hemicellulose molecules, whereas cellulase enzymes break down cellulose molecules’ linkages. Table 3 lists the yield of bioethanol production with different types of microorganisms. Furthermore, cellulases are a family of enzymes that collaborate to break down cellulose into glucose monomers. *Endoglucanases, exoglucanases, cellobiohydrolases*, and *β-glucoside* enzymes all play a role in this enzymatic pathway. *Endoglucanase* properly cuts cellulose chains at the ends, whereas endoglucanase hydrolyzes intramolecular *β-1,4-glucosidic* bonds of cellulose chains randomly to form a new chain end. Hemicellulose consists of heteropolysaccharides with complicated polymer. Hemicellulose are divided into five-carbon sugars, such as xylose and arabinose; and six-carbon sugars, such as galactose, glucose, and mannose. Firstly, The pretreatment approach eliminates lignin, followed by an enzymatic saccharification process to make simple sugars from lignocellulosic biomass [75,76].

On the other hand, the chemical structure of the polymer is quite complicated, and to degrade the polymer of hemicelluloses, such as xylan, we required multiple enzymes. *Trichoderma* spp., *Penicillium* spp., *Talaromyces* spp., *Aspergillus* spp., and *Bacillus* spp. are among the fungus and bacteria that generate xylan degrading enzymes. *Endoxylanase, exoxylanase, β-xylosidase, α-arabinofuranosidase, α-glucuronidase, acetyl xylan esterase*, and ferulic acid are among the enzymes involved in the enzymatic hydrolysis of xylan [77,78].

Bacteria and fungi can generate cellulases, such as Thermomonospora, Cytophaga, Sporocytophaga, Cellulomonas, Erwinia, and Clostridium, as well as bacteria, including *Thermobifida*, *Caldicellulosiruptor*, *Butyrivibrio*, *Bacillus*, *Fibrobacter, Acetovibrio*, *Microbispora*, *Cellulomonas*, *Ruminococcus*, *Streptomyces*, *Microbispora*, and *Bacteroides*. It demonstrated promise in the hydrolysis of various types of lignocellulosic biomass [79,80]. The product of enzymatic hydrolysis is glucose. Glucose fermentation by rich industrial host stains would increase bioethanol yield through the greatness of bioethanol production. *Zymomonas mobilis* and *Saccharomyces cerevisiae* are two such strains. Nevertheless, they cannot utilize pentoses. The most common pentose fermenting yeasts are *Candida shehatae, Scheffersomyces stipitisa,* and *Pachysolen tannophilus* [81,82].

White-rot, brown-rot, and soft-rot fungi are the most commonly applied for lignocellulosic biomass hydrolysis. It depends on the type of lignocellulosic biomass; each fungus has a different method of action. Because these kinds of fungi produce numerous lignin-degrading enzymes, such as lignin peroxidases, manganese-dependent peroxidases, polyphenol oxidases, laccases, white-rot, and soft-rot fungi effectively degrade lignin. Nevertheless, the brown-rot fungus is interested mainly in cellulose components [83]. Wei et al. studied corn stover’s effect with ferric chloride as catalyzed through various additives by enzymatic saccharification. The FeCl_3_-catalyzed dimethyl sulfoxide pretreatment may improve biomass hydrolysis. The extracts of hemicellulose and lignin were 100% and 36.4%, respectively. The maximum yield of glucose is up to 90.2% by Cellic CTec2 cellulase. Tween 80, as an additive, is a potential surfactant to decrease cellulase dosage and reduced hydrolysis time [84].

**Table 3 polymers-13-02886-t003:** Highest-yield bioethanol production with different types of microorganisms.

Feedstock	Enzymatic Hydrolysis			Type Enzyme for Hydrolysis	Bioethanol Promotor	Fermentation			Type	Y_Ethanol_, %	Reference
	T (°C)	pH	t (h)			T (°C)	pH	t (h)			
Fringe (*Chionanthus retusus*)	45	5	48	*Novozymes*	*Saccharomyces cerevisiae*	37	5	48	SSF	81	[74]
Sugarcane bagasse	50	4.8	90	*Cellic CTec2 cellulase*	*C. tropicalis*	20–28	NA	NA	NA	91	[85]
Mixed sawdust	50	4.8	NA	*Cellic CTec2 cellulase*	*Saccharomyces cerevisiae ATCC* *7754*	28	NA	24	NA	80	[86]
Rice straw	45	4.8	72	*Cellic CTec2 (VCNI0013) and Cellic HTec2 (VHN00002)*	*Saccharomyces cerevisiae CCUG 53310*	37	5	48	DSSF	98.7	[87]
Corn and corn stover	50	4.8	48	*α-amylase and glucoamylase*	*Saccharomyces cerevisiae GIM2.213*	30	5.5	96	SSF	99.3 (g/L)	[88]
Willow (*Salix viminalis* W)	40	4.8	30	*NA*	*Saccharomyces cerevisiae*	30	NA	24	NA	65	[89]
Sugarcane bagasse	NA	4.8	96	*Novozyme 188 (cellubiase of Aspergillus niger)*	*Zymomonas mobilis*	30	5	15	NA	84	[90]
Triticale straw	37	5	72	*Spezyme^®^ CP, Optiflow™ RC 2.0, Accellerase^®^ 1500 and Celluclast^®^ 1.5*	*Saccharomyces cerevisiae Ethanol Red^®^*	37	5	144	SSF	84.7	[91]
Poplar wood	50	5.5	72	*Cellic^®^CTec3* *Novozymes*	*Saccharomyces cerevisiae YRH400*	37	NA	48	SSF	68	[92]
Fresh softwood, *Picea abies*, free from bark	40	4.8	96	*Novozym 188*	*Saccharomyces cerevisiae*	37	5	72	SSF	65	[93]

Y_Ethanol_ = yield of ethanol, %.

### 4.2. Acid Hydrolysis

Acid hydrolysis aims to derive fermentable sugars that are used in the fermentation process for bioethanol production. Table 4 shows the acid hydrolysis process with different types of biomasses for bioethanol production. The fermentable sugars are primarily kept in crystalline hemicellulose structures. A polymer is made up of glucose, xylose, and other sugars [94]. These sugars can be recovered more than 80% from hemicellulose by using acid hydrolysis [95]. The first step of acid hydrolysis is to disintegrate the matrix structure of untreated biomass into cellulose, hemicellulose, and lignin [96]. The subsequent hydrolysis is to convert polysaccharides of treated biomass into monosaccharides. The monosaccharides are utilized as a feedstock for the fermentation process for making bioethanol. The most common acids used for hydrolysis are sulfuric acid, phosphoric acid, and hydrochloric acid [97,98,99]. The advantage of acid hydrolysis for producing sugar in the bioethanol process is that the acid can easily access the cell wall. The rate of acid hydrolysis is faster than that of other processes [100]. 

Nevertheless, the drawbacks of using acid are adding the cost for neutralization in the subsequent process and causing an environmental pollutant. The neutralization process in the acid hydrolysis process is essential before fermentation to prevent a toxic environment for microorganisms. The acid hydrolysis process can be divided into two methods: dilute acid concentration (0.1%) at a higher temperature (>200 °C) and concentrated acid (30–70%) at a lower temperature (<50 °C) [101]. Dilute acid is widely used for industry because of its high reaction rate; moreover, it will increase cellulose hydrolysis. In addition, this process will utilize fewer acid concentrations, but the higher temperature of this process will consume more energy. On the other hand, concentrated acid hydrolysis uses less energy because it is conducted at a lower temperature. The concentrated sulfuric acid hydrolysis used obtained the maximum yield of sugar from woody biomass [102]. However, the higher acid concentration will inhibit enzymatic activity (furfural, 5-hydroxymethylfurfural) and increase equipment corrosion [103]. 

## 5. Fermentation Process

Five fermentation strategies can be applied for bioethanol production: (1) For cell recycling batch fermentation (CRBF), the main objective of the CRBF process is to adapt yeast to become inhibitors. Furthermore, the CRBF process increases yeast performance. It can decline reaction time and minor carbon deviation for cell production. (2) For simultaneous saccharification and fermentation (SSF), the enzymatic hydrolysis and fermentation are applied to release sugar to produce ethanol. Nevertheless, the disadvantages of the SSF process are the low performance between the ideal temperatures of the enzyme and yeast. (3) Separated hydrolysis fermentation (SHF), hydrolysis, and fermentation are conducted separately to produce bioethanol. (4) For semi-simultaneous saccharification and fermentation (SScF), the short pre-hydrolytic is applied before SSF process. The bioethanol yield would be increased slightly than that of conventional methods of SSF. (5) Consolidated Bioprocessing (CBP) involves the decomposition of resistive biomass substrates into solubilized sugars, as well as a metabolic intervention to guide the metabolic flow toward specific products with high yield and titer [104,105,106,107,108,109]. Table 5 indicates the advantages and disadvantages of various fermentation processes.

Kongkeitkajorn et al. studied bioethanol production by SHF and *S. cerevisiae* TISTR 5339 as a microorganism with the total bioethanol up to 44.7 g/L after 24 h of cultivation. Additionally, xylose uptake declined during the fermentation, with 32.9% of the total amount. The maximum concentration of 31.3 g/L achieved by using *S. shehatae* ATTC 22984 was directly connected to glucose consumption, which was substantially lower than that obtained by using *S. cerevisiae*. The inactivation of ethanol production after 48 h showed that there might be a product inhibitory impact on *S. shehatae* growth [110]. Furthermore, the different types of fermentation processes and microorganisms are shown in Table 6. 

The hydrolysis and SSF process are carried out simultaneously in the same units. Many process variables affect SSF activity, including solid loading, enzyme loading, inoculum size, temperature, pH, inhibitors, and additional medium components [111]. These processes have advantages, such as enhancing the yield, reducing the need for enzymes, reducing the total processing time, and increasing the hydrolysis rate by generating sugars that restrict cellulase activity at the same time. Because glucose is directly eliminated from the medium, it removes the need for sterile conditions. The disadvantage of this method is the incompatibility of hydrolysis and fermentation temperatures, necessitating the introduction of thermotolerant bacteria [112,113]. *Clostridium acetobutylicum*, *P. stipitis* NCIM 3498, *S. cerevisiae*, *Issatchenkia orientalis*, *Thermoanaerobacter ethanolicus*, and *Karl Marxianus* are some of the thermotolerant/thermophilic fermenting strains [114].

**Table 4 polymers-13-02886-t004:** Acid hydrolysis with various biomass feedstock.

Feedstock	Pretreatment	Acid Hydrolysis	Acid Concentration	Acid Hydrolysis		Microorganism	Microorganism Concentration	Y_Sugar_	Y_Ethanol_	Reference
				T (°C)	t (h)					
Cassava	NA	H_2_SO_4_	0.58 M	100	0.5	*Saccharomyces cerevisiae*	5%	28.18%	14.7%	[97]
Waste potatoes	Ultrasonic	HCl	2.1%	40	48	*Saccharomyces cerevisiae*	19.2 g/L	NA	65.8 g/L	[98]
Sugarcane bagasse	NA	H_2_SO_4_	0.1 M	120	2	NA	NA	452.27 mg/g	NA	[99]
Citronella biomass	NA	H_2_SO_4_	0.1 M	120	2	NA	NA	487.50 mg/g	NA	[99]
Softwoods	NA	H_2_SO_4_	95%	40	0.67	NA	NA	46.2%	NA	[102]
Potato peel	NA	H_2_SO_4_	5%	90	1.5	*Saccharomyces cerevisiae*	NA	65 g/L	6.45 g/L	[115]
Sago pith waste	Microwave	H_2_SO_4_	1 M	120	0.016	*Saccharomyces cerevisiae*	5% (*v*/*v*)	0.67 g/g	0.31 g/g	[116]
Potato tuber	NA	HCl	1 M	-	1	*Saccharomyces cerevisiae*	60 g/L	94%	31 g/L	[117]
Sago	NA	H_2_SO_4_	1.5 M	90	1.5	NA	NA	0.6234 g/g	NA	[118]
Rice straw	Mechanical	H_2_SO_4_	2 M	90	1	NA	NA	9.71 g/L	0.013%	[119]
Waste papers	Thermal	H_2_SO_4_	1%	96.31	0.344	*Saccharomyces cerevisiae*	NA	79.65% (*w*/*v*)	16.5%	[120]
Lignocellulosic	NA	H_2_SO_4_	0.3–4% 0.3–6%	130–220 190–240	0.0167–1 0.0167–0.167	*Saccharomyces cerevisiae*	NA	55–60%	90%	[121]
Wheat straw	NA	H_2_SO_4_	2%	180	0.167	*Pichia stipitis* NCIM 3498	10%	19.32%	5.29%	[122]
Seaweed *Ulva rigida*	Thermal	H_2_SO_4_	4%	NA	1	*Pachysolen tannophilus*	5%	34 ±0.25 mg/mL	0.37 g/g	[123]
Spruce wood	NA	H_2_SO_4_	0.05 M	200	1	NA	NA	124.54 mg/g	NA	[26]
Beech wood	NA	H_2_SO_4_	0.05 M	200	0.67	NA	NA	148 mg/g	NA	[26]
Spruce wood	NA	H_2_SO_4_	70%	80	8	*Saccharomyces cerevisiae*	1%	70%	74.3%	[124]
Birch wood	NA	H_2_SO_4_	70%	80	8	*Saccharomyces cerevisiae*	1%	70%	64.7%	[124]

Y_Ethanol_ = yield of ethanol. Y_Sugar_ = yield of sugar.

**Table 5 polymers-13-02886-t005:** The advantages and disadvantages of different types of fermentation and microorganism.

Feedstock	Pretreatment	Fermentation	Type of Microorganism	Advantages	Disadvantages	Reference
Corn stover	EDA	SScF	*S. cerevisiae* SyBE005	Enhanced ethanol concentration	High enzyme load	[125]
Sugarcane bagasse	PMS combined with AD	SScF	*Saccharomyces cerevisiae* SHY07-1	High titer and yield ethanol produced No inhibitor was produced	Low glucose consumption efficiency might develop into carbon deficiency and have a poor impact on cell viability	[126]
Hardwoods Vineyard waste	HPAC Autohydrolysis	SSF	*Saccharomyces cerevisiae*	Short hydrolysis duration High ethanol productivity High yield ethanol produced	Long duration of saccharification The product acts as a feedback inhibitor The optimal conditions for hydrolysis and fermentation are still to be identified Not appropriate for fermentation at high temperatures	[74,127]
Waste Paper	Acidic alkaline pretreatment	SHF	*Saccharomyces cerevisiae*	A high rate of sugar to ethanol conversion	Unable to ferment the pentose produced by hemicellulose hydrolysis.	[128,129]
Soybean residue	Thermal	SHF	*Saccharomyces cerevisiae* KCCM 1129 adapted to galactose	Increase the total yields of ethanol fermentation.	Fermentation has been interrupted due to modifications in the fermentation stage	[130]
Microalgae	Thermal	SHF	*Z. mobilis* ATCC 29191	High efficiency of glucose utilization with dilute acid Higher glucose concentrations can be produced with 2% sulfuric acid	Low glucose concentration with dilute acid, with 2% sulfuric acid Producing some inhibitory compounds that cause poor glucose utilization and lower ethanol yield	[131]
Beechwood	Thermal	CBP	*C. thermocellum* strain ATCC 31924	Full lactate inhibition Enhanced ethanol production.	The significant potential for cellulose degradation and of ethanol Require long periods of fermentation	[132]

EDA = ethylenediamine; PMS = potassium peroxymonosulfate; AD = alkaline deacetylation; HPAC = hydrogen peroxide/acetic acid.

Guilherme et al. stated that, when using sugarcane bagasse treated with acid-alkali and *S. cerevisiae* PE-2, the theoretical yield of ethanol generated was 92% after 18 h [133]. Zheng et al. also stated that using *Saccharomyces cerevisiae* with unwashed corncob residues at 10% (*w/w*) substrate concentration of tea-seed cake at 15 FPU/g-cellulose produced the maximum ethanol production of 93.3% [134]. Simultaneous saccharification and co-fermentation (SScF) is a process that includes simultaneous hydrolysis and fermentation of C_5_ (pentose) and C_6_ (hexose) sugars in the same unit. When substantial participation of the pentose is discovered following hydrolysis, this method is proposed. The advantages of this process are faster production rates, a higher bioethanol yield, and a lower risk of contamination. The disadvantages are needed to have high enzyme loads [135]. The fermenting organisms generate the enzymes in the CBP process. Cost-effectiveness and energy efficiency are the advantages, but a limitation of appropriate microorganisms and difficulties controlling the process are the disadvantages [109,136]. Park et al. investigated the ethanol production, enhanced to 18.2 g/L (106.1% of total glucose loading) at 168 h, using recombinant *E. coli* (K011). This finding implied that the SScF utilizing recombinant *E. coli* (K011) yielded more than 100% of the predicted maximal ethanol production based only on glucan. The results indicated that recombinant *E. coli* (K011) would efficiently consume glucose and xylose and convert them to ethanol [137].

**Table 6 polymers-13-02886-t006:** Ethanol production using various types of fermentation and microorganism.

Raw Material	Microorganism	Method of Pretreatment	Temperature and Reaction Time			Type	Y_Ethanol_, %	Reference
			Pretreatment	Hydrolysis	Fermentation			
Corn stover	*Two Saccharomyces cerevisiae*	Liquid hot water	180 °C; 10 min	50 °C; 72 h	35 °C, 96 h	SScF	3.57% (*w*/*v*)	[138]
Sugarcane bagasse	*Two Saccharomyces cerevisiae*	Liquid hot water	180 °C; 10 min	50 °C; 72 h	35 °C, 96 h	SScF	3.29% (*w*/*v*)	[138]
Wheat straw	*S. cerevisiae strain Ethanol*	Organosolv	180 °C; 40 min	50 °C; 72 h	35 °C, NA	SSF	67.24%	[139]
Cotton stalk	*S. cerevisiae*	Microwave assisted alkali	100 °C; 10 min	50 °C; 96 h	30 °C, 96 h	SSF	41.28%	[140]
Cotton stalk	*S. cerevisiae + P. tannophilus*	Microwave assisted alkali	100 °C; 10 min	50 °C; 96 h	30 °C, 96 h	SSF	56.47%	[140]
Cotton stalk	*S. cerevisiae + P. tannophilus*	Biological pretreatment with LZ-K2	37 °C; 7 days	50 °C; 96 h	30 °C, 96 h	SSF	54.73%	[140]
Sugarcane bagasse	*Aspergillus niger*	Diluted phosphoric acid pretreatment	120 °C; 20 min	50 °C; 72 h	30 °C, 24 h	CRBF	47.00%	[141]
Chlamydomonas Mexicana biomass	*Chlamydomonas mexicana*	Sonication and enzymatic hydrolysis	50 °C; 15 min	50 °C; 24 h	30 °C, NA	SSF	50.00%	[142]
Corn stover	*Saccharomyces cerevisiae*	Co-solvent	150 °C; 25 min	50 °C; 18 h	37 °C, 120 h	SSF	89.20%	[143]
Hydrolysate of rice straw	*S. cerevisiae*	Hydrothermal	NA	NA	30 °C, 24 h	SSF	46.00%	[144]
Hydrolysate of corncob	*Pichia guilliermondii*	NA	NA	50 °C; 72 h	30 °C, 18 h	SHF	44.40%	[145]
Food waste	*Zymomonas mobilis*	Biological pretreatment	NA	50 °C; 6 h	30 °C, 44 h	SHF	50.00%	[146]
Sugarcane bagasse	*Saccharomyces cerevisiae*	Steam explosion	195 °C; 7.5 min	50 °C; 96 h	35 °C, 24 h	SHF	24.50%	[147]
Ethanol-extracted cane bagasse	*Saccharomyces cerevisiae*	Steam explosion	195 °C; 7.5 min	50 °C; 96 h	35 °C, 24 h	SHF	29.60%	[147]
Sugarcane bagasse	*Saccharomyces cerevisiae*	Steam explosion	195 °C; 7.5 min	50 °C; 96 h	35 °C, 48 h	SSF	28.00%	[147]
Ethanol-extracted cane bagasse	*Saccharomyces cerevisiae*	Steam explosion	195 °C; 7.5 min	50 °C; 96 h	35 °C, 48 h	SSF	30.80%	[147]
Cashew apple bagasse (15% CAB-OH)	*Kluyveromyces marxianus ATCC 36907*	Acidic alkaline pretreatment	121 °C; 15 min	40 °C; 12 h	40 °C, 72 h	SSF	92.7%	[148]

Saha et al. reported fermentation with *Saccharomyces cerevisiae* (Baker’s yeast) at 25 °C, pH = 7 to get 0.333 mg/L ethanol. Moreover, 0.133 mg of ethanol can be produced from 1 g of Pteris, where the conversion reducing sugar to ethanol is 20% approximately in the optimum reaction condition [149]. Joannis-Cassan et al. presented the industrial alcoholic fermentation carried out by *Saccharomyces cerevisiae* in the multi-stage batch and fed-batch fermentation, producing 15.2% (*v/v*) ethanol in 53 h without residual sucrose and with ethanol productivity of 2.3 g L h^−1^ [150].

## 6. Separation of Bioethanol

Pretreatment, hydrolysis, fermentation, and separation are the four main processes in synthesizing bioethanol from lignocellulosic biomass. The separation process is a crucial part of the bioethanol manufacturing process because it needs more energy. Due to the more dilute alcohol concentrations encountered in the second-generation processes, distillation and dehydration account for around 20 to 40% of the total energy consumption for first-generation bioethanol production. For example, alcohol composition varies between 0.75 and 5.0 wt% ethanol after fermentation [151]. The normal ethanol composition in the water-ethanol solution is 89.4 mol% at 78.2 °C because of the azeotrope. The different types of separation processes of bioethanol include azeotropic distillation, extractive distillation, dehydration by vacuum distillation, membrane process, adsorption process, and chemical dehydration process [152]. Distillation technologies such as azeotropic and extractive distillation have relatively high energy costs. However, they are still the preferred approach for large-scale bioethanol fuel generation despite this main disadvantage. Hajinezhad et al. reported that the distillation unit consumed about 70% of the total energy requirements for the procedure to produce 99.5 wt.% bioethanol [153]. 

Figure 4 shows the separation process of bioethanol via extractive distillation. Furthermore, the conventional azeotropic and extractive distillations are conducted with two columns. The first stage is ordinary distillation, also known as the pre-concentration stage, the percentage of bioethanol concentration is up to 92.4–94 wt.%. In addition, the second stage of dehydration of ethanol is applied to get a higher concentration than conventional azeotropic distillation. Advanced process intensification and integration techniques, thermally connected distillation columns, dividing-wall columns, heat-integrated distillation columns, or cyclic distillation are new ways to solve energy-intensive distillation [154,155,156,157]. 

The separation of bioethanol via membrane process is conducted by mass transfer of liquid and gas streams. The anhydrous ethanol was produced by using hyperfiltration (reverse osmosis), gas/vapor permeation, and pervaporation [152]. Some chemical plants with tens to hundreds of square meters of the membrane have used tubular and plate-frame modules. The spotlight of industrial pervaporation research has recently shifted to bioethanol dehydration. Furthermore, several thousand meters of the membrane will be required for the bioethanol separation process. It needed more cost-effective module designs. Ube created hollow fiber polyimide membranes in Japan for this application. Pervaporation is adapted by other firms, using spiral-wound module technology [158].

Another approach for obtaining anhydrous bioethanol is through pervaporation. A semipermeable membrane is used in this technique. This method effectively separates azeotropes and close boiling liquids because the separation of the membrane is not dependent on the liquid-vapor equilibrium. Khalid et al. studied how to separate ethanol by adding a small external agent with a high boiling point [159]. Figure 5 shows the process flow diagram of the bioethanol separation process with a simultaneous distillation/pervaporation facility for ethanol recovery from fermenters. The range concentration of bioethanol through dehydration and distillation process for 95–99.5%. Furthermore, molecular sieve is one of the conventional and commercial methods used to separate bioethanol [160]. On the other hand, azeotropic ethanol is produced by using a stripper column, followed by a rectification column; the ethanol as a feedstock is fed into the pervaporation process. The pervaporation process typically operates in the temperature range of 105–130 °C, with the vapor pressure of the feed stream up to 2–4 bar to optimize the vapor pressure difference and pressure ratio across the membrane [158].

Chong et al. gave a techno-economic evaluation of a biorefinery configuration, using macroalgae cellulosic residue (MCR), using Aspen Plus V10. A total of 15,833.3 kg/h of MCR generated 7626 kg/h of anhydrous bioethanol and 3372 kg/h of fertilizer. The design achieved positive energy saving by performing Heat Exchanges Networks Synthesis and process optimization and could reach a net energy ratio of 0.53. It also showed economic viability with a minimum selling price for anhydrous ethanol of $0.54/kg [161]. Techno-economic analysis of ethanol production from lignocellulosic biomass was investigated by a previous study [162]. Angili et al. used the life-cycle assessments (LCA) to assess bioethanol production. The pretreatment technique was the most carefully considered because pretreatment plays such a crucial role in subsequent processes. The updated pretreatment procedures result in environmental savings. Nevertheless, advanced pretreatment methods and input and output optimization in bioethanol production can help reduce environmental impacts caused by increasing acidification, eutrophication, and photochemical oxidant on our planet [163].

Ethanol production at high substrate loadings could effectively decrease the equipment size and reduce the consumption of water and the cost of ethanol separation. Lu et al. showed that the bioethanol concentration reached 66.5 g L^−1^, and the bioethanol yield of 0.133 g g^−1^ was achieved by using 21% substrate [164]. The high substrate concentration is a limiting factor in the heat and mass transfer of the reaction system but also increases the content of inhibitor in the hydrolysate and eventually reduces the conversion efficiency of ethanol. Zheng et al. obtained the highest ethanol yield of 0.257 mg/g sugarcane bagasse by using a solid-liquid ratio of 1:20 (*w*/*v*) [165].

Recycling technique is vital for bioethanol production from lignocellulosic biomass, including recycling chemical wastes (toxicants accumulation), recycling wastewater, reusing high-cost additives, and reusing byproducts production. All recycling techniques should be applied in pretreatment, enzymatic hydrolysis, fermentation, and distillation to decrease the production cost; reduce the consumption of chemicals, cellulase, and yeast; and increase economic feasibility [166]. The use of agricultural residues for bioethanol production greatly depends on the availability of raw materials and the proper design of a flexible multi-feedstock facility. Durtle et al. studied multiple coffee crops’ residues (stems, pulp, and mucilage) to show that the production costs and the CO_2_ emissions obtained were 0.5 $USD/L and 1.29 Kg CO_2_/L, respectively. The mass balances were used to calculate the blue and gray water footprints. The blue water footprint was calculated by using water utilized in LHW (liquid hot water) pretreatment, washing, and reaction. Outflow water and water contained in streams (solids, vinasse, and sludge) were used to calculate the gray water footprint. The huge facility uses 3.364 million tons of fresh water per year and generates 119,000 tons of gray water annually. On the other hand, without recirculation requires 2.87 times more water (1.170 million tons per year of fresh water) [30].

## 7. Conversion of Bioethanol into Bio-PET

PET is used as a raw material in the production of fibers (65%) and packaging products (35%). In addition, PET as a raw material of packaging products is mainly used as plastic bottles (76%), containers (11%), and films (13%) [1,167]. PET has outstanding chemical and physical features for various applications, including gas barrier capabilities, low diffusivity, superior mechanical and thermomechanical properties, extremely inert material, clarity, and satisfactory process operation [168,169,170]. PET feedstocks are commercially produced from petroleum-based materials. As a result, these raw materials contribute to the unsustainable nature of PET-based products. In addition, the short-term use of PET-based products causes serious environmental issues, such as slow degradation and abundance of untreated PET-based product wastes [7].

Furthermore, the production of bio-PET should be to find sustainable raw materials with lower cost in the process production. It helps maintain an ecological balance, which is critical for the long-term survival of our planet [168,171,172]. Using biomass as a raw material for producing ethylene glycol (EG), terephthalic acid (TPA), and dimethyl terephthalate (DMT) is one of the alternative solutions to solve these environmental problems and to improve the sustainability of PET. Table 7 lists the industrial-scale of renewable bio-based PET products. PET can be produced in two different ways. The first method involves TPA with EG. The second process is used DMT and EG by transesterification reaction [2].

Currently, commercially produced bio-PET is partial bio-PET produced by polymerizing biomass-based EG and petroleum-based TPA. About 30% of the carbons constituting PET are derived from biomass, which is called bio-PET 30. The worldwide production capacity of bio-PET 30 in 2016 was 94,800 tons [173]. Shen et al. calculated greenhouse gas (GHG) emissions of 2.34 kg-CO_2_/kg-bio-PET 30, using corn-derived EG. As GHG emissions of petro-PET were reported to be 3.36 kg-CO_2_/kg-petro-PET, bio-PET 30 showed a 30.4% greater reduction of the GHG emissions than petro-PET [174].

Bio-PET as a final product can be synthesized from bio-EG and bio-TPA by the polymerization reaction. The polymerization of PET is divided into two steps. In the first step, bio-TPA and bio-EG are reacted to produce a monomer unit in the esterification process. Nitrogen is needed for this autocatalytic reaction. The second step is forming the chain of PET in the vacuum condition with the temperature reaction up to 280 °C [167,175,176].

**Table 7 polymers-13-02886-t007:** Industrial-scale of renewable bio-based EG and PET product and precursors.

Company	Country	Raw Materials	Products	Reference
Gevo Inc, (Englewood)	USA	Corn	Bio p-xylene	[176]
India Glycols Limited	India	Molasses	Bio-EG	[177]
JBF Industries Ltd.	Brazil	Sugarcane	Bio-EG	[7]
Greencol Taiwan Corporation (GTC)	Taiwan	Sugarcane	Bio-EG	[7,178]
Toyota Tsusho Corporation	Japan	Sugarcane	Bio-EG, Bio-PET	[178]
Teijin Ltd.	Japan	Sugarcane	Bio-EG, Bio-PET	[179]
Japane Future Polyesters	Japan	NA	Bio-PET	[1]
Coca-Cola—Gevo Venture	USA	NA	Bio-PET	[1]
Futura Polyesters	India	NA	Bio-PET	[180]
Far Eastern New Century Corporation	Taiwan	Agriculture Waste	Bio-PET, Bio-MEG	[181]
Indorama (Guangdong IVL PET Polymer)	China	Biomass	Bio-PET, Bio MEG	[182]

MEG: mono ethylene glycol.

Recently, EG can be made with various raw materials such as ethanol, glycerol, sorbitol, sugars, and cellulose. Bioethanol derived from biomass is commercially used in the manufacture of EG in Figure 6. However, compared to EG production from renewable feedstocks, the production of PTA from biomass remains is limited because of the abundance of raw material (p-xylene) from the crude oil refining process. Due to the sustainable issue, the greener new reaction routes of PTA production renewable resources are increased. Bio-based p-xylene can now be made from various renewable resources, including isobutanol, 5-hydroxymethylfurfural (HMF) with ethylene, limonene, bio-ethylene, isoprene with acrylic acid, and furfural [7,183].

The sequential chemical reaction processes are used to produce bio-EG. The reaction of the aliphatic compound of bio-EG might be synthesized from the hydrolysis of ethylene oxide. It could be achieved by oxidization. The oxidation of bio-ethylene is produced during glucose fermentation, followed by dehydration [184,185]. Furthermore, the production of bio-ethylene by catalytic dehydration of bioethanol is one the most sustainable and cost-effective ways [186,187]. The various catalysts are used in the dehydration process: alumina, silica, zeolites, clays, and phosphoric acid, with maximum selectivity and conversion up to 99.9% and 100%, respectively [188,189] (Table 8). The dehydration is conducted in a fixed or fluidized bed reactor with a catalyst under the vapor phase. The reaction can be achieved by isothermal or adiabatic with fixed-bed reactor.

Meanwhile, it is commonly applied in adiabatic with fluidized bed reactor [190]. In addition, bio-ethylene could be blended with CO_2_, argon, oxygen, nitrogen, or methane. In a tubular catalytic reactor, the dilute gas mixture is fed into the reactor. The chemical reaction produces bio-ethylene oxide under extremely exothermic conditions with temperature and steam drum pressure. The bio-ethylene oxide is scrubbed with water. On the other hand, the byproduct of CO_2_ would be returned and eliminated to the reactor loop through bio-ethylene oxide’s flow into the glycol reactor. In contrast, the ethylene glycol is produced via a chemical reaction with water then the multi-effect evaporator is applied to remove the water compound [16].

### Bio-Terephthalic Acid Production

TPA is mainly produced via the oxidation of fossil feedstocks such as para-xylene [191]. The reaction routes of bio-TPA are considered by using various feedstocks, such as isobutanol, HMF plus ethylene, limonene, and lignin. In the first two routes, para-xylene is converted into TPA, using the Amoco^®^ process [183]. Bio-TPA is formed by bio-paraxylene, which is performed with bio-isobutanol as a precursor in the reaction. Gevo Inc developed the process of bio-paraxylene production, using bio-isobutanol from the fermented corn feedstock. The bio-paraxylene process consists of three reactions: bio-isobutanol dehydration, bio-isobutylene dimerization, and bio-diisobutylene dehydrocyclization in Figure 6. The dehydration reaction was conducted at a temperature from 300–350 °C, the pressure at 60–200 psig, and a weight hourly space velocity (WHSV) of 1–20 h^−1^. Furthermore, the reaction is carried out by using BASF AL-3996 as a dehydration catalyst. The isobutylene yield was up to 95%. The bio-isobutylene products from the first reaction step were fed into the dimerization reactor to synthesize bio-diisobutylene (2,4,4-trimethylpentanes, 2,5-dimethylhexenes, or 2.5-dimethylhexadienes). The acidic condition of the catalyst is required for the removal of water from the feedstock. The catalyst for this reaction must also be tolerant of the water produced by the reaction. In addition, before entering the subsequent process, the water from the reaction product must be removed. The reactor’s operating pressure determines the selection of the water separation process at this stage. When the reaction is conducted at 0–30 psig, a gas-liquid separator can be utilized to separate it. Nevertheless, the liquid-liquid separator can be used when the reactor pressure is between 30 and 100 psig. The bio-isobutylene product from the previous process is put into the fixed bed reactor to be converted into bio-diisobutylene through dimerization or oligomerization reaction. The dimerization is carried out in a fixed bed reactor that contains HZSM-5 as a catalyst and can function properly in a liquid state. The reactor’s operating temperature is kept at 170 °C, the pressure at 170 psig, and the WHSV at 20 h^−1^. The main product dimerization consists of at least 50% of 2,2,4-trimethylpentanes. Due to this low conversion reaction, some unreacted feedstock must be recycled into a dimerization reactor to improve the conversion to 99%. The dimerization reaction occurred with an oligomerization catalyst. The bio-diisobutylene isomers were converted into bio-paraxylene by dehydrocyclization reaction in the fixed-bed reactor with BASF D-1145E catalyst. The selectivity of the dehydrocyclization reaction is up to 75%. Unreacted isobutylene from the dimerization reactor is recycled to dilute the feedstock of the dehydrocyclization reactor to improve conversion and selectivity. Before converting to terephthalic acid, bio-paraxylene from this reaction must be purified to meet the purity requirement. Separation units, such as simulated moving bed chromatography, fractional crystallization, or fractional distillation, can improve bio-paraxylene purity. High purity bio-paraxylene from the previous reaction is oxidized with air or oxygen at 80–270 °C to synthesize the purified terephthalic acid. The oxidation reaction is carried out with a metal catalyst (manganese, cobalt, or nickel) and chemicals. Acetic acid and bromide compounds (hydrogen bromide, bromine, or tetrabromoethane) are added to the reactor as the solvent and additional oxidation agents. Tibbetts et al. reported an efficient elevated-pressure catalytic oxidative process (2.5 mol% Co(NO_3_)_2_, 2.5 mol% MnBr_2_, air (30 bar), 125 °C, acetic acid, 6 h) to oxidize p-cymene into crystalline white terephthalic acid (TA) in ~70% yield [192].

**Table 8 polymers-13-02886-t008:** Dehydration of ethanol to ethylene by using different catalysts.

Type of Catalyst	Maximum Ethylene Selectivity, %	Ethanol Conversion, %	Temperature Reaction, °C	Stability	Reference
Spherical catalyst with ϒ-Al_2_O_3_	99.6–100	99.5–100	350–450	Very Stable	[193]
TPA-MCM-41	99.9	98	300	Very Stable	[194]
Spherical silica particle (SSP) and alumina-silica composite (Al-SSP)	99	98	400	Very Stable	[195]
Ti-deZSM-5	88	96	280	Stable	[196]
Zeolite HZSM-5 modified by 0.5% La-2%P	99.9	100	240–280	Very Stable	[197]
Al_2_O_3_–MgO	97	97–100	450	Very Stable	[198]
Co-Cr/SAPO-34 (silicoaluminophosphate)	99.4	99.15	400	Very Stable	[199]
TiO_2_/ϒ-Al_2_O_3_	99.4	100	360–500	Very Stable	[200]
STA-MCM-41	99.9	99	250	Stable	[201]
ϒ-Al_2_O_3_	94	99	350–450	Stable	[202]
Zeolite ZSM-5	99	100	240	Very Stable	[15]

On the other hand, bio-TPA can be derived from furan, for instance, 2,5-furan dicarboxylic acid (FDCA). The Diels–Alder (DA) reaction is the essential chemical reaction in bio-TPA production from furan derivatives. At the beginning of this process, maleic anhydride is formed by oxidizing and dehydrating furfural and then reacting with furan to form a DA adduct. When the DA adduct is dehydrated, phthalic anhydride is formed and then transformed to bio-TPA, using dipotassium phthalate and phthalic acid. Avantium, the leading producer of bio-based FDCA, reported on another fascinating bio-TPA synthesis process utilizing DA. Hydrogenation converts hydroxymethylfurfural (HMF) to dimethyl furan, a key precursor of FDCA (DMF). Following various stages, such as cyclization with ethylene by DA and dehydration, DMF is changed to p-xylene, then turned to bio-TPA [203,204]. Furthermore, the production of TPA from renewable biomass feedstock (e.g., lignin) has been attracting research interest [205,206]. Settle et al. reported TPA synthesis from biomass-derived aromatic compounds via isomerization [207]. Bai et al. reported that the yield of TPA obtained was 58.7% from lignin-based phenolic acids, which included hydrogenation, demethoxylation, and carboxylation reactions [208]. Song et al. reported the production of TPA from corn stover lignin [209]. They use a three-step strategy to produce TPA from lignin-derived monomer mixtures to produce corn-stover-derived lignin oil with a supported molybdenum catalyst. The overall yields of TPA based on the lignin content of corn stover could reach 15.5 wt%. This bio-based PET is identical to petrochemical PET and can be processed by injection molding, blow molding, and extrusion. Nevertheless, it is not biodegradable. Salvador et al. provided an interesting article that described microbial degradation of PET due to the action of microbial polyester hydrolase, which was regarded as a key option for PET recycling [1,210].

## 8. Conclusions

The physical, chemical, physical-chemical, and biological pretreatments can be used to handle lignocellulose before converting it into bioethanol in the fermentation process. *Saccharomyces cerevisiae* is the type of microorganism that could achieve the highest yield, up to 99.3%, with rice straw as a feedstock. The separation process of bioethanol production is challenging. Because of the azeotrope condition between water and ethanol, the maximum ethanol purity is 89.4% at 78.2 °C. There are several technologies to handle the azeotropic condition, for instance, azeotropic distillation, extractive distillation, and membrane separation. However, the energy consumption is very high. The promising technology that can produce higher purity of bioethanol is the pervaporation method. The purity of bioethanol can be achieved at 95–99.5 wt%. The industrial production scale of bio-PET is conducted by collaborating with other companies, for instance, Coca-Cola with Gevo Venture. The companies recently used one bio-based reactant for producing bio-PET. Coca-Cola achieves it by using 100% bio-based EG. The main challenge in the future is using 100% bio-based reactants to make 100% bio-PET. In conclusion, the importance of selecting pretreatment methods should be emphasized in handling lignocellulose. In addition, extensive research on separation bioethanol to cope with the lower purity of ethanol must be conducted.

## Figures and Tables

**Figure 1 polymers-13-02886-f001:**
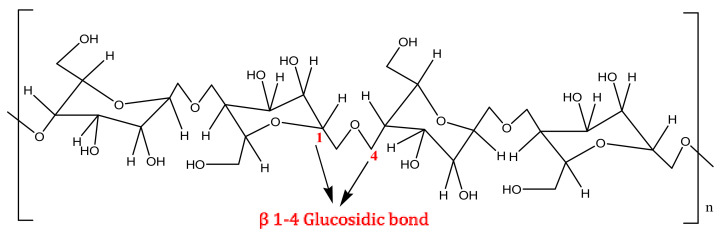
The chemical structure of cellulose.

**Figure 2 polymers-13-02886-f002:**
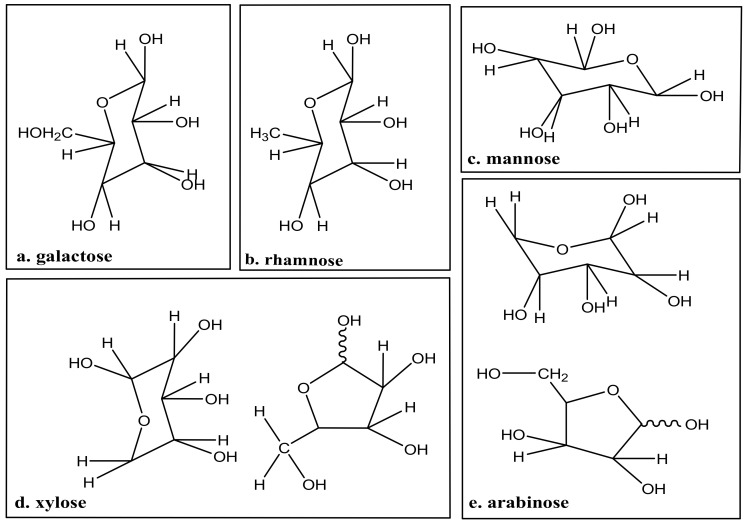
Chemical models of sugar in hemicellulose.

**Figure 3 polymers-13-02886-f003:**
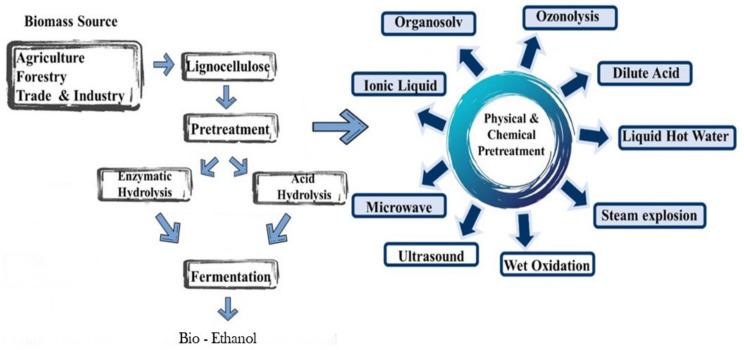
Schematic conversion of the lignocellulose to produce bioethanol.

**Figure 4 polymers-13-02886-f004:**
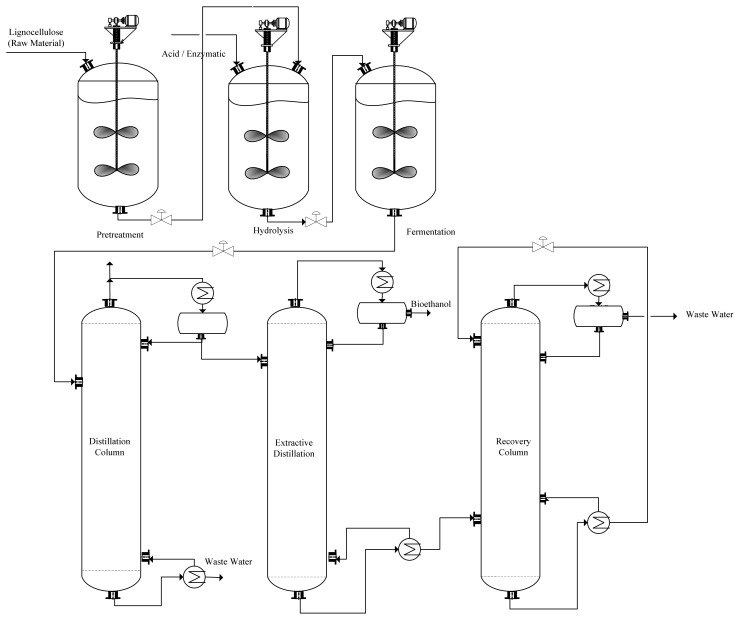
The separation process of bioethanol by extractive distillation.

**Figure 5 polymers-13-02886-f005:**
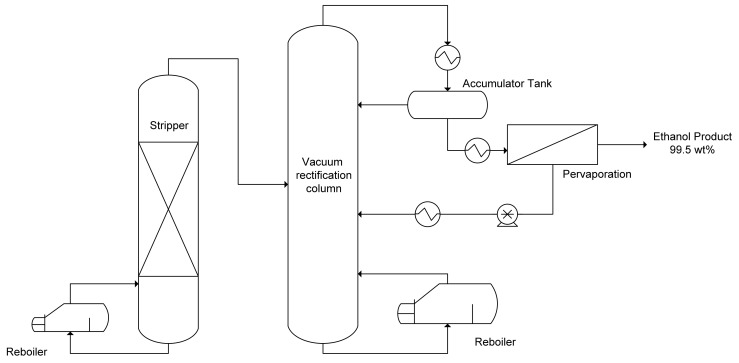
A simultaneous distillation/pervaporation facility for ethanol recovery from fermenters. (Reproduced with permission from Baker, R.W, Membrane Technology and Application; published by John Wiley and Sons, 2004).

**Figure 6 polymers-13-02886-f006:**
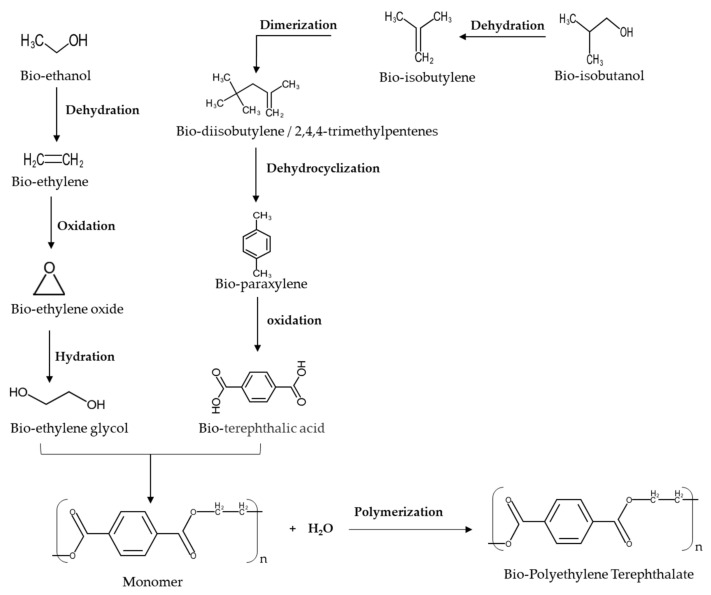
Reaction routes to Bio-PET.

## Data Availability

Not applicable.

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
