# Peer review of "Conversion of Lignocellulose for Bioethanol Production, Applied in Bio-Polyethylene Terephthalate"

_polymers, 2021, doi:10.3390/polym13172886_

Round 1

Reviewer 1 Report

The manuscript authored by Damayanti and his associates presents a relevant overview of the studies dealing with the conversion of lignocellulose for bioethanol production and with chemical reactions of bioethanol to bio-ethylene and bio-terephthalate acid to produce bio-polyethylene terephthalate.

Throughout the whole text, there are no logical transitions between the sentences, i.e. they are not connected with each other. It is laudable of the present review to have about 65% references over the recent 5 years, but the analysis of the references has been made very poor. A lot of data were mixed, from which the authors drew the conclusions.

Comments and questions:

  1. Lines 41-49. Introduction. The information on the structures of lignocellulose and lignin is better to move from the Introduction to Section 2 or rule out at all. I advise the authors to avoid repetitions.
  2. Figure 3 is called as “Schematic conversion of the lignocellulose ethanol process”, but in addition to ethanol, it illustrates the synthesis of “Chemical derivatives” via the “Conversion and Synthesis” stage after enzymatic hydrolysis of lignocellulose. It should be made clear as to which specific processes are meant here. And why the derivatives are adduced separately as the product of conversion and synthesis following enzymatic hydrolysis, so is ethanol given separately and the schematic ends with it? It is in disagreement with the manuscript title. And the whole paper is concerned with the synthesis of derivatives from ethanol.  
  3. Figures 4 and 5 are not informative, I advise that these are removed.
  4. Table 1. The column “Microorganism” wrongly lists hydrolytic enzyme preparations (Cellic CTec2 cellulase) in some lines rather than the producer of ethanol . One more column “Enzymes” should be added, and they should not be confused with producers of ethanol.
  5. Details on the feedstock and its pretreatment method are missing in Table 3, in order that conclusions can be made.
  6. I advise the authors to cite a larger and more qualitative critical review on the potentiality of lignocellulose for over 200 value-added compounds, including that for ethanol and polyethylene terephthalate: F.H. Isikgor, C.R. Becer. Lignocellulosic biomass: a sustainable platform for the production of bio-based chemicals and polymers. Polym. Chem., 2015, 6, 4497. DOI: 10.1039/c5py00263j.

Reviewer 2 Report

  • The authors have to edit the manuscript type from “Article” to “Review”.
  • There are many typing mistakes such as missed spaces, additional letters, etc., for examples see lines: 79, 93, 124, 147, 254, etc.
  • In line 116 edit the mistake in (4-O-5 and s).
  • In line 189, is better to write the numbers in scientific form.
  • In line 192, the celsius degree symbol is incorrect, it should be edited in the whole manuscript.
  • It should be a space =between the numerical values and units.
  • The introduction section is very limited. The authors have to improve the introduction part, some of the following references may be discussed and incorporated into this section:

https://doi.org/10.1007/s40995-020-00948-7 ; https://doi.org/10.1007/s00289-020-03301-5 ; https://doi.org/10.2174/1573413716666200310121947 ; https://doi.org/10.1007/s00339-018-1890-0

Reviewer 3 Report

Major Revision:

This article deals with the properties and conversion of lignocellulose into bioethanol, the separation process of bioethanol, the production of bio-terephthalic acid, and the chemical reaction from bioethanol to bio-polyethylene terephthalate. The chemical reactions of bioethanol to bio-ethylene and bio-terephthalate acid to produce bio-polyethylene terephthalate were investigated using both theoretical and experimental issues.

The paper meets the aim and the scope, as well as, the high academic standards of the ‘Polymers’ Journal. However, the following specific improvements should be made, before accepting the paper for possible publication to the Journal.

General comments:

  • Please do not split the words at the end of the line. Please revise this grammar error throughout the manuscript. For example in lines: 2, 4, 14, 16, 20, 23, 36, 40, 41, 43, 44, 47, 51, 61, 62, 63, 65
  • The authors need a native English-speaker to thoroughly revise the grammar and syntax of this manuscript. Please carefully proof-read spell check to eliminate grammatical and syntax errors. For example:
    • Line 50-52: Please rewrite this sentence as the meaning of it is not clear.
    • Line 53-54: This sentence does not connect with the previous one. Please be careful with the use of linkers.
    • Line 60-62: These sentences are not clear and the coherence is lost. Please revise it accordingly using the appropriate linker.

Abstract:

  • Line 15-16:Due to environmental issues, the raw materials of polyethylene terephthalate can be produced from lignocellulose.” This sentence is misleading. PET can be produced from biomass lignocellulose regardless of the environmental issues. Please revise this sentence accordingly.
  • Line 16-17: “Converting lignocellulose into bioethanol has many challenges and potential due to its sustainability.” The term sustainability is too general. Please be more specific regarding the challenges and potential of converting lignocellulose to bioethanol.
  • Line 22-23: “The conversion of lignocellulose provides a solution to solve the environmental problems…” This sentence is too general and confusing. Please avoid generic statements that could be criticized.
  • Please include the knowledge gap and the major conclusions of the paper.

  1. Introduction
  • Since the idea of reviewing the development from lignocellulose to bio–polyethylene terephthalate, is not a new one, at the end of this section, the authors should provide a clear and concise understanding of the primary contribution of their manuscript.

  1. Properties of Lignocellulose
  • Line 96 - 97: “The cellulose, hemicellulose, and cellulose are broken down by…” The term “cellulose” referred twice in this sentence. Please revise it accordingly.

  1. Conversion of lignocellulose into bio-chemical
  • Please revise the title of this chapter to “Conversion of lignocellulose into bio-chemicals” or “Conversion of lignocellulose into bio-chemical substance”.
  • A comparative table of the different types of pretreatment processes, including conditions, rates, cost and other comparable key features, is strongly recommended to be added.

  1. Lignocellulose Hydrolysis
  • Line 268-271: Please revise this sentence, especially the figures given in brackets, as it is not clear if these figures referred to the cost of the bioethanol process or to electricity consumed.
  • Line 371 (Table 2): For comparison purposes, all yields and concentration should be given in the same units.
  • An overview table of advantage and disadvantage of enzymatic and acid hydrolysis is strongly recommended to be added.

  1. Fermentation Process
  • Line 373 (Table 3): Please rewrite the advantages and disadvantages in each case in bullets.

In 2nd line: Please explain why the production of succinic acid is an advantage for the fermentation process.

  1. Separation of Bioethanol
  • Line 468: “The bioethanol was [144].” This sentence has not been completed. Please revise it accordingly.
  • Line 482-484: The point of this sentence is not clear. Do you mean the biomass production have adverse impact on acidification, eutrophication, and photochemical oxidant formation and positive effect on Global warming? Please revise this sentence.
  • Line 504: Please provide further details about the water problem investigated in the study [28].

  1. Conversion of Bioethanol into Bio–polyethylene terephthalate
  • Line 542: Please provide the meaning of the acronym “MEG” in the Notation table.

  1. Conclusions
  • Conclusion must be contain basic findings of the review also with the knowledge gap the paper would be filling. Additionally, this section is poor; more discussion is required.
  • A Discussion part should be included in order to describe in detail the findings of the study. Please write this section according to the template provided by Polymers Journal.
  • Kindly provide strong recommendations for future researches.

References

  • The context of this paper should be enriched with more studies published in the Polymers Journal.

Round 2

Reviewer 1 Report

Typos in the new version of the article:

  1. Line 288 The word "saccharomyces" must be capitalized
  2. Table 4 The word "Seaweed" is underlined.